# LOGICAL VIEW ON FAIRNESS OF A BINARY CLASSIFICATION TASK

## ABSTRACT

Ethical, Interpretable/Explainable, and Responsible AI are an active area of research and important social initiative. Vendors offer solutions. For instance, Microsoft compiled a platform, Responsible AI. Within the context, challenges of algorithmic fairness and trustworthiness of machine learning are paramount. Furthermore, several authors argue that the emergence of algorithmically infused societies necessitates innovative approaches to measuring feasible information, e.g., collecting data shall follow a trustworthy social theory. In this paper, we show that this approach is heuristic at best.

We prove that, with no regards to data, fairness and trustworthiness are algorithmically undecidable for a basic machine learning task, the binary classification. Therefore, even the approach based on not only improving but fully solving the three usually assumed issues – the insufficient quality of measurements, the complex consequences of (mis)measurements, and the limits of existing social theories – is only heuristics. We show that, effectively, the fairness of a classifier is not even a (version of bias-variance) trade-off inasmuch as it is a logical phenomenon. Namely, we reveal a language $L$ and an $L-$theory $T$ for binary classification task such that the very notion of loss is not expressible in the first-order logic formula in $L$.

## 1   INTRODUCTION

Ethical, Interpretable/Explainable, and Responsible AI are an active area of research and important social initiative. Vendors offer solutions. For instance, Microsoft compiled a platform, Responsible AI. Within the context, challenges of algorithmic fairness and trustworthiness of machine learning are paramount. Furthermore, several authors argue that the emergence of algorithmically infused societies necessitates innovative approaches to measuring feasible information, e.g., collecting data shall follow a trustworthy social theory [3]. Difficulties, associated with such approach, can be found in [7]. Moreover, in this paper, we show that this approach is heuristic at best.

We prove that, with no regards to data, fairness and trustworthiness are algorithmically undecidable for a binary classification task (cf. [4], [5]). Therefore, even the approach based on not only improving but fully solving the three usually assumed issues – the insufficient quality of measurements, the complex consequences of (mis)measurements, and the limits of existing social theories – is only heuristic. We prove that, effectively, the fairness of a binary classifier is not even a trade-off (e.g., a version of bias-variance/complexity etc.) inasmuch as it is a logical phenomenon. Namely, we reveal a language $L$ and $L-$theory $T$ for binary classification task such that the very notion of loss is not expressible in the first-order logic $L$-formula.

Note that the essence of a "mass view" approach is that unlike in a traditional machine learning context, we are not making any assumptions on nature of a classifier loss, other than it should provide a way to compare two (potentially different) classifiers. Under this very broad perspective, it turns out that, in a natural model, the loss of a classifier is inexpressible as a first-order logic expression (cf. Appendix for the definitions). Without loss of generality, it follows that any feasible definition of fairness for machine learning classification task is undecidable. Indeed, one has to assume that two classifiers have to be comparable in their performance characteristics in the first place. If the latter is not expressible, then one cannot achieve a sensible conclusion on fairness. By the same token, since all derived heuristics such as transparency, interpretability and trust, must include a notion of fairness,

the undecidability result immediately generalizes to these concepts (thus complementing results of [4]).

More specifically, we present an almost surely decidable model where the classifier loss is not expressible. Thus, undecidability of a classifier loss is not necessarily associated with undecidability of the model. However, if we utilize yet another view of binary classifiers over an infinite domain, the class, viewed as a lower bounded lattice, is (first-order logic) undecidable.

Throughout the paper, we consider the natural generalization of the binary classifier for the infinite domain.

Our goal is to introduce a purely logical view on loss for a binary classifier on an infinite domain. It is achieved by introducing a general notion of classifier loss based on an observation that any natural loss is a first-order formula in a suitable structure. The latter has a theory $T$ represented by a tuple $\langle L, M \rangle$ where $L$ is a language and $M$ is model for $L$.

Next, we will show that the resulting first-order theory $T$ admits an extension $RG_{ext}$ on random graph structure such that a notion that a graph has an equal number of connected and unconnected nodes is not expressible in the $RG_{ext}$-first-order logic. The binary classifiers' structure is isomorphic (with probability 1 to $T$. Therefore, if a first-order sentence in one theory is deducible (i.e., can be proved) in that theory, the corresponding sentence is deducible in another.

Then, for two given classifiers, assuming that the classifier loss is a first-order logic formula, $C1 \equiv \langle D, L_{c1} \rangle$ and $C2 \equiv \langle D, L_{c2} \rangle$, we can construct a first-order expression $L_{c1} - L_{c2} = 0$ which is equivalent to an expression that the two classifiers have the same number of connected and unconnected nodes which leads to contradiction. This effectively means that any loss function is not expressible in the $RG_{ext}$-first-order logic. The rest of the paper is dealing with the proof of these statements. It is interesting to compare this with a general undecidability of identities for wide class of functions in [1].

We conclude with discussion of losses expressible in second and higher logic theory and immediate implications for adopting them on fairness and interpretability (the extended version of the paper contains more information on each of these topics).

## 1.1 NOTATIONS AND DEFINITIONS

We will try to make this paper self-sufficient and provide all necessary references to the reader who would like to invest more time into the mathematical foundations of machine learning and interpretablity. We would need some information from model theory. We assume that the reader is familiar with the concepts of domain, classifier, and loss as well as the standard body of statistics and probability theory normally used in a supervised machine learning. Notations are natural; $\mathbb{N}$ denotes the set of natural numbers, $\mathbb{Z}$ stands for integers, and $\mathbb{R}$ denotes reals. $\mathbb{R}_+$ would be positive reals. $L$ or $l$ normally stands for a loss unless it is a space which is then defined explicitly. $S \sim D$ means a sample from a distribution $D$; contextually, $D$ can stand for a domain.
In general, we assume an infinite countable domain.
Traditionally, given a hypotheses space $H$ and domain $Z$, loss $l$ is a non-negative real function $l : H \times Z \to \mathbb{R}_+$.

We denote $L_D(h)$ a standard expected loss of a binary classifier $h \in H$ over domain $X$ where $H$ is a hypotheses space, with respect to a probability distribution $D$; by definition: $L_D(h) = E_{z \sim D}[l_{0-1}(h, z)]$, and, since for 0-1 loss Z ranges over pairs,

$$l_{0-1}(h, (x,y)) = \begin{cases} 0 & \text{if} \quad h(x) = y \\ 1 & \text{if} \quad h(x) \neq y \end{cases}.$$

We also need some definitions from model theory and logic. A filter $\alpha$ on the set of natural numbers $N$ is a collection of sets of natural numbers obeying the following axioms:
1) If $E \subset F \subset \mathbb{N}$ and $E \in \alpha$ then $F \in \alpha$;
2) If $E \in \alpha$ and $F \in \alpha$ then $E \cap F \in \alpha$;
3) Empty set $\emptyset \notin \alpha$.

An ultrafilter $\alpha$ on the natural numbers is a filter which obeys an additional axiom:
If $E \subset \mathbb{N}$ then exactly one $E$ or $\mathbb{N} \backslash E \in \alpha$.
A non-principal ultrafilter $\alpha$ is an ultrafilter that obeys one additional axiom yet:
4) No finite set belongs to $\alpha$.

It easy to see that non-principal ultrafilters exist. We can start with the filter of cofinite (i.e., the complements of finite sets in $\mathbb{N}$), and applying Zorn lemma [3] to embed the filter into an ultrafilter. We call an ultrafilter principal if it is not non-principal. One can prove that every non-principal ultrafilter is of the form $\{E \subset \mathbb{N} | n \in E\}$, where $n$ is a natural number.
One fundamental property of binary classifier $\mu : 2^N \to \{0, 1\}$, if it maps $\sigma$-large sets to 1, and $\sigma$-small sets to 0, is that it is a finitely additive probability measure. Moreover, every finitely additive probability measure has this form.

We also are going to use an important tool from model theory that is called ultraproduct $\prod_{i \in I} L_i / U$ where $L_i$ are some algebraic structures indexed by $i \in I$, and $U$ is an ultrafilter.
Algebraic operations on the ultraproduct are defined the same way as in a Cartesian product of $L_i$ and the ultraproduct is a quotient set with respect to relation $\diamond$ which is defined as follows: $x \diamond y$ iff $\{i \in I | x_i = y_i\} \in U$ .

Since we need some notions from lattice theory, we remind the fundamentals of it. Lattice is a partially ordered set in which any two elements have a supremum and an infimum. $Z_l, l \in \mathbb{N}$ is a lattice $(Z_l, \leq)$ if we set $a \wedge b = inf\{a, b\}$ and $a \vee b = sup\{a, b\}$.

Then $Z_l$ is an algebra and a lattice. For a lattice $L$, we put $\uparrow a = \{b \in L | b \geq a\}$ and $\downarrow a = \{b \in L | b \leq a\}$.

If we fix the domain, then a multi-class classifier, as a structure, is defined (by the lattice isomorphism) by the ordered set of values in $Z_l$.
$L_D(A)$ denotes a standard 0-1 loss for a binary classifier $A$.

Finally, we need to touch decidability. Intuitively, it is the situation in first-order logic theory when we can find an effective algorithm that decides whether a well-defined formula for the theory is true (in particular, it does not loop indefinitely – the very reason why halting problem is undecidable). The first-order logic is a natural setting for the context, because the second order (and, in general, a higher order) logic only leads to difficulties in any formalization or contemporary understanding of interpretability. The celebrated Gödel incompleteness theorem, for instance, deals with the second order formulas.

These definitions are sufficient for our purposes. We need only point out that these fundamental notions normally are essential for a non-standard analysis. However, we deal with a standard context herein and just use them in some proofs.

## 1.2 RANDOM GRAPH CLASSIFIERS

There is one interesting class of classifiers, similar in nature to binary classifiers, such that its first-order theory is almost for sure decidable (in the sense of probability). This class is associated with random graphs. We would call those the random (binary) classifiers.
The class is defined as follows. Assume that domain is enumerated (countable). Consider a graph with vertex set $\mathbb{N}$ of all natural numbers. Pick a "random" binary classifier $A$ which is the one that decides on every scoring randomly, with probability $\frac{1}{2}$. [1]. Let $RG$ be the resulting structure.

To decide whether there is an edge between $x$ and $y$ (with $x \neq$ y), we check values $A(x)$ and $A(y)$. If they coincide, then, by definition, there is an edge, otherwise there is not. Being viewed as a structure, the class of binary classifiers is a bounded lattice in case of a finite domain, and a left bounded lattice in case of infinite domain.

---

[1]In fact, the very same proof works if, within the context, we assume a ratio $\frac{p}{q}$ where $p$ and $q$ are integer numbers such that $0 \leq \frac{p}{q} \leq 1$.

The class of all graphs built this way forms the class of random graphs. We show next that all these graphs are isomorphic (as graphs) with probability 1 (i.e., almost for sure). Namely, consider two disjoint finite subsets of $N$, $X$ and $Y$, with respected sizes $n$ and $m$. Consider now an element $x \in \mathbb{N}$ that is joined to every element of $X$ and no element in $Y$; we would call that a property $P$. Given any vertex $x$, the probability it does not have the property $P$ above is $p = 1 - 2^{-(n+m)}$. So, for $m$ different vertices, the probability that none of them has the desired property is $p^m$. Clearly, this would converge to zero if $n$ is going to infinity. Thus, at least one $x \in N$ will have the property, with probability 1. Taking into account the fact that there are only countably many disjoint pairs $(X, Y)$ of finite sets in $\mathbb{N} \times \mathbb{N}$, with probability 1, for each pair $(X, Y)$ we can find a vertex that $x$ is joined to every vertex in $X$ and to no vertex in $Y$. Let $\mathcal{P}$ be that property.

Denote the resulting structure by $RG_{ext}$ defined by the property $\mathcal{P}$. Any binary classifier $C \equiv \langle D, L_c \rangle$ may be viewed as an element in $RG$ where each element of the domain is represented by a pair of vertices in $RG$. Loss of the classifier $C$ is defined accordingly and becomes a first-order expression in the theory $T \equiv L_{0Ext}$-theory where $L_{0Ext}$ is the extension of basic logical language $L_0$ extended with the property $\mathcal{P}$.

From that point, we can simply repeat the arguments from [2], Chapter IV.23, pp.645-646 and deduce that the first order theory of random graphs is almost surely decidable, so the first order logic theory of binary classifiers, associated with random graphs, is almost surely decidable too.

In particular, we can consider the extension $RG_{ext}$ for $RG$ with additional property that with probability 1, for every pair $(A, B)$ of disjoint subsets in our domain, there is a vertex $v$ such that $v$ joined to every vertex in $A$ and no vertex in $B$.

Q.E.D.

Namely, we have the following

**Proposition 1.1.** *The theory $L_{0Ext}$ of random graph classifiers $RG_{ext}$ is almost surely decidable.*

For completeness, we provide the proof based on an extension in the appendix.

## 2    MAIN RESULT

### 2.1    INEXPRESSIBILITY OF FAIRNESS

Let theory $T$ be (as above) the $L_{0Ext}$-theory of binary classifiers based on structure $RG_{ext}$. Despite of the fact that the theory is almost surely decidable, once cannot use any first-order expression to compare two binary classifiers. Namely, the following theorem holds:

**Theorem 2.1.** *Two binary classifiers cannot be compared fairly using any first-order logic expression in $T$.*

*Proof.* Assume that the classifier loss is expressible in $L_{0Ext}$-first-order logic of $T$. By Marker's theorem for almost sure theory of graphs (e.g., cf. [2], p. 646), the theory $T$ obeys a zero-one law which states that for any $L_{0Ext}$-sentence $\phi$ the probability $p_N(\phi)$ tends to zero or tends to 1 as $N \to \inf$. On the other hand, one can see that the probability that a random graph contains $r(\frac{N}{r})$ edges tends to $1/r$ as N tends to infinity for any integer $r > 1$.

Translating this to classifiers in $RG_{Ext}$, consider any non-trivial classifier. Its loss function then would tend to one or zero as we enumerate the elements of domain. Thus, we witness a contradiction with the fact that the limit is between zero and one. Q.E.D. □

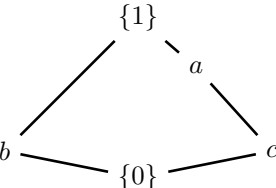

Figure 1: Finite Lattice Diagram representation

## 3 UNDECIDABILITY

**Theorem 3.1.** *Given a domain $D$, the class of binary classifiers over $D$ (viewed as lower bounded lattices) is (first-order logic) undecidable.*[2]

*Proof.*

The main idea is to show that the class is not closed under ultraproducts (or Cartesian products). Then, the class is not axiomatizable. As a consequence, being viewed as a structure, it is not decidable. We first consider the case of finite domain. Strictly speaking, not every binary classifier, viewed as a structure, is a lattice. However, if it is, the immediate observation is that, up to a lattice isomorphism, such binary classifier is bounded. Namely, for a lattice $L$, $a \in L$ consider set $\uparrow (a) = \{b \in L | b \geq a\}$. A lattice homomorphism $h : M \to L$ is lower bounded if for any $a \in L$, set $h_{-1}(\uparrow (a))$ is either empty or has the least element. A lattice is lower bounded, if any homomorphism from a finitely generated free lattice is lower bounded. Dual definitions are established for the upper bounded lattices.

If lattice is lower and upper bounded, then it is called bounded. Any binary classifier is lower bounded as a lattice if it is representable as a lattice. It is obvious for finite classifiers, since we can view classifier as a lexicographically ordered sequence of pairs $(x_i, y_i), i \in N$ and thus it contains 0 (the lowest bound) . Given a finite domain, it also contains 1; any homomorphism preserves 0s and 1s. For an infinite domain $D$, we assume it can be countably enumerated (thus, the order of the product $D \times D$ is lexicographical). Then, any classifier on such domain is lower bounded, since $D$, as a lattice, would contain 0.

Any finite lattice $L$ can be identified as a binary classifier $l : L \times L \to \{0, 1\}$ that is associated with the lattice diagram in the following manner. The lattice is represented, up to isomorphism, by its diagram which is a directed graph $G = G(V, E)$ such that for every $v \in V \subset L$ there is no $x$ such that $a > x > c$ for vertices $a$ and $c$ with edge $e(a, c)$ (covering property). The direction of the $e$ is pointing towards a smaller element. Then we simply set classifier $l$ value to 1 for the pair $(a, c)$. If a pair $(x, y)$ is not in the diagram, we set the value to zero. It is not difficult to check that the diagram can be restored from the classifier only one way (cf. figure 1 for the process, l(a,c) →1 and l(c,a) →0).

---

[2](Informally, there is no efficient algorithm that decides whether a well-formed formula in the first order logic theory of binary classifiers is true. Therefore, if we try to find an "interpretable" explanation for a phenomenon being explained by a model, then, in general, that explanation may not be possible to formulate in first order logic terms. One may argue that this reduces the very notion of interpretability to heuristics).

We need the following result from [6].

**Theorem 3.2.** *There are finite lower bounded lattices $L_i, i \in I$ for Cauchy filter $D$ such that $\prod_{i \in I} L_i / D$ is not lower bounded.*

The theorem is stated there for just bounded lattices; however, it is valid for lower bounded lattices as well - for completeness, we present the proof in the appendix. To prove our result, theorem 3.1, we only need to notice that, according to the theorem 3.2, the product is a lattice, but it is no longer a binary classifier, that is, there are finite (lower bounded) binary classifiers such that their ultraproduct is not a binary classifier.

Q.E.D.

## 4  CONCLUSION

An elementary class (or axiomatizable class) consists of all structures satisfying a fixed first-order theory. There are a few limitations of expressiveness of elementary classes. Theorem 3.1 seems to be a reason to move to the second-order logic; however, with higher expressiveness, and some inheritance of the first-order logic properties, there are important negative effects, associated with completeness and compactness theorems. More practical approach is to use the first-order logic.

**Discussion**   One may ask a natural question whether the second-order and higher order logic is more suitable for the modeling the machine learning tasks for formalization of fairness, because the first-order logic has quantifiers' scope "reduced" to elements, rather than sets. Clearly, binary classification task is only useful tool if we can verify the model at the individual level (e.g., the target "malignant/non-malignant" tumor in medicine). Moreover, we would like our formalism be as simple as possible, that is, our expression of boolean target shall refer to individual prediction rather than subset of those, because the problem of defining and selecting the subsets is likely (and, logically, will be) more complex than our task in the first place, at least, for majority of practical problems. By the same token, interpretability requires its scope to be at the individual elements.

Once more point is permissible here for computer science settings. The mass view approach is fundamentally different from adopted (un)decidability patterns of computability theory (e.g., Rice theorem about semantic properties of a program). It is a viewpoint inspired by model theory.

In addition, one important consideration is due to formal verification. Many modern verifiers and theorem provers use a higher order logic since they employ type theory. However, the verification and automatic proving are yet other research domains that are not being discussed in this paper.

### ACKNOWLEDGMENTS

The author thanks Prof. V. Pestov for his extremely useful comments. In particular, he pointed out that the phenomenon of inexpressibility can be associated with nonmeasurability of the classifier (the standard model just assumes Borel measurability). That prompts an additional question about the classifier such as what its Bayes loss is which we do not address in this paper since it is out of scope for the mass view approach on classification that does not make any assumption on measurability.

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

# A APPENDIX

## A.1 PROOF OF PROPOSITION 1.1

**Theorem A.1.** $RG_{ext}$ *is almost surely decidable.*

*Proof.* This is a completion of the proof sketched in the main text. We aim to construct an extension for the first-order theory such that it is almost for sure decidable. In the main text, we proved that, with probability 1, there is a countably many disjoint pairs $(X, Y)$ of finite sets, for every such pair one can find $x \in X$ that is joined to every vertex in $X$ and to no vertex in $B$. That extension property formalization is as follows:

(i) $\forall x \forall y \quad x \sim y \to y \sim x$;
(ii) $\forall x \quad \neg(x \sim x)$;
(iii) $\Phi_{n,m}$ for $n, m \geq 0$

where $\Phi_{n,m}$ is the formula:

$\forall x_1 ... x_n \quad \forall y_1 ... y_m \wedge_{i=1}^n \wedge_{j=1}^m x_i \neq y_j \to \exists z((\wedge_{i=1}^n x_i \sim z) \wedge (\wedge_{i=1}^m \neg(y_i \sim z))).$

Consider the theory $T$ for graph with the extension property (i)-(iii) holds for any pair of disjoint finite set of vertices. By the argumentation above in the main text, with probability 1, random classifiers are models of $T$.

Moreover, any two countable models of $T$ are isomorphic. The isomorphism can be constructed as follows. Induction. Given model $M_1$ and $M_2$, and enumerations $a_0, a_1, ...$ of vertices in $M_1$ and $b_0, b_1, ...$ of vertices in $M_2$, let set $f(a_0) = b_0$. For $a_1$, if $a_1$ is joined to $a_0$, find a vertex that is joined to $b_0$, and in similar way, if not. These are property $\Phi_{1,0}$ and $\Phi_{0,1}$ correspondingly. Selecting alternatively an image for the first $a_i$ that does not yet have an image, and preimage for the first $b_j$ that does not yet have preimage, and a regular case of $\Phi_{i,j}$, we continue by induction.
The theory $T$ is complete. Furthermore, there is an algorithm that decides whether any formula or its negation are true in very model of $T$. That is a simple corollary from the previous considerations. To prove completeness, assume that the opposite is true. Then, by compactness theorem, there are two countable models $M_1, M_2$ such that a formula is true in $M_1$, but its negation is true in $M_2$. This would mean that there are two non-isomorphic models of $T$ which is not the case. Similarly, to prove that there is a deciding algorithm for $T$, for any formula, one would search the proofs for it or its negation. By completeness theorem (Gödel, Henkin), syntactic and semantic consequence are equivalent for first-order logic, so we will find eventually a proof for the formula or its negation.

To prove almost for sure decidability for random graphs, we only need to notice that $T$ provides the proof for a formula $\phi$ or its negation. The proofs are finite, so the proof can use only finitely many statements $\Phi_{n,m}$. Thus, there exists $m \in \mathbb{N}$ such that if $M \models \Phi_{m,m}$, then $M \models \phi$. As we saw before, the probability of $M \models \Phi_{m,m}$ tends to 1 when $m \to 0$, so is the probability of $M \models \phi$. This means that the first order theory of random graphs is almost surely decidable.
Q.E.D.

$\square$

## A.2 PROOF OF THEOREM 3.2

*Proof.*

That statement is the main result of [6]. Following the very construction of [6], we need only to build an ascending chain of elements in the free lattice that its image under natural embedding saves the order of elements.

Next, given a domain, we can identify classifier with an embedding into an inverse limit of finite bounded lattices. More specifically, let $X$ will be a finite set with at least free elements (that is not a real restriction for binary classifier case, because logically multinomial classifier can be split into a

sequence of binary ones), and let $F(X)$ be a free lattice generated by X. It is known [8] that $F(X)$ embeds into an inverse limit of finite bounded lattices.

Specifically, there are finite bounded lattices, $L_i, i < \omega$, and surjective lattice homomorphisms $\pi_{ij} : L_j \to L_i, i \le j < \omega$ such that $\pi_{ii} = Id_{L_i}$ and $\pi_{ik} = \pi_{ij}\pi_{jk}$ for all $i \le j \le k < \omega$, and an embedding $\phi : F(X) \to \prod_{i<\omega} L_i$ for which $\pi_i\phi = \pi_{ij}\pi_j\phi$ with any $i \le j < \omega$ where $\pi_k : \prod_{i<\omega} L_i \to L_k$ is the canonical projection for any $n < \omega$.
Fix a filter $\Phi$ over $\omega$, e.g., the filter of all cofinite subsets of $\omega$. Then the map $\xi : F(X) \to \prod_{i<\omega} L_i/\Phi; \xi : a \to \phi(a)/\Phi$ is a lattice homomorphism, by definition. In fact, it is a lattice embedding. Namely, suppose $a \ne b$ in $F(X)$, then we have $\pi_i\phi(a) \ne \pi_i\phi(b)$ for some $i < \omega$. For the inverse limit, it means that $\pi_n\phi(a) \ne \pi_n\phi(b)$ for all $n \ge i$. Immediately, $\xi(a)/\Phi \ne \xi(b)/\Phi$.
Next, consider $t \in F(X)$ and an ascending chain $\{a_i, i < \omega\}$ in $F(X)$. Then $\{\pi_n\phi(a_n)|n < \omega\}/\Phi \le \xi(t)$ if and only if $a_n \le t$ for all $n < \omega$.
We will prove now that homomorphism $\xi$ is not lower bounded. It is known (cf. [8], Example 1.24, Whitman) that $F(X)$ contains an infinite ascending chain $\{a_n|n < \omega\}$ which does not have a least upper bound.
Consider $\bar{a} = (\pi_n\phi(a_n)|n < \omega)/\Phi$. Since $1_{F(X)} \in \xi^{-1}(\uparrow \bar{a})$, it follows that the last set is non-empty. Therefore, if we choose an arbitrary element $t$ from the set, we have $\xi(t) \ge \uparrow \bar{a}$. As we saw before, it means we have $t \ge a_n$ for all $n \le \omega$. Since $t$ is not a least upper bound for the sequence $\{a_n|n < \omega\}$, there exists $t_1$ such that $t > t_1 \ge a_n$ for all $n < \omega$. As usual, it means that $\xi(t_1) \ge \uparrow \bar{a}$ and thus $t_1 \in \xi^{-1}(\uparrow \bar{a})$. Hence the set $\xi^{-1}(\uparrow \bar{a})$ does not have a least element, i.e., $\xi$ is not lower bounded.

Q.E.D.

One variation of the proof for binary classifiers is based on the following consideration. For finite and countably infinite domains, binary classifier is a distributive lattice over $N$ if we set $\{a \vee b, a \wedge b\} = \{a, b\}$ $\forall a, b$. As is known, for very partial set (poset) $C$ there exists a unique, up to isomorphism, free completed distributed lattice $L$ over $C$ with embedding $\phi : C \to L$ such that for every distributed lattice $M$, and a monotone function $f : C \to M$ there exists a unique homomorphism $f_\phi$ that the following diagram is commutative:

$$
\begin{array}{ccc}
C & \xrightarrow{\varphi} & L \\
\downarrow{\scriptstyle f} & & \downarrow{\scriptstyle f_\phi} \\
M & \xrightarrow{id} & M
\end{array}
$$

This allows us to uniquely identify a classifier (as a distributive lattice) with a free distributed lattice embedding; more precisely, with its image under the embedding $\phi$. Since $L$ is a free lattice generated by $C$, it, in turn, embeds into inverse limit of free finite bounded lattices [6].

Next, it is sufficient to prove that there are finite lower bounded lattices the embedding above, that are identified as binary classifiers, that for a non-principal ultrafilter $D$ (e.g., $\omega$), the ultraproduct $\prod_{i \in I} L_i/D$ is not lower bounded.
Then, we can find an infinite chain in the embedding directly, and the rest of the proof is as above.

## A.3    FORMAL DEFINITION FOR LANGUAGE $L$

Let $L$ be a language (an extension of a basic formal logical language, $L_0$).

**Definition**    The set of $L-$terms is the smallest set $L_t$ such that contains all constant symbols of $L$, all variables, and if $t_1, t_2, ..., t_n$ are in $L_t$ then for any n-ary function symbol $f$, $f(t_1, t_2, ..., t_n)$ is also in $L_t$. Set $L_a$ of atomic formulas are represented by the properties:
(1) if $t_1$ and $t_2$ are terms then $t_1 = t_2$ is in $L_a$, and
(2) the corresponding n-ary function symbols are also in $L_a$.

In other words, the set of all formulas in $L$ (expressions, sentences - herein, we use these interchangeably) is the smallest set containing all atomic formulas and closed under logical connectives $\vee$, $\wedge$, $\neg$, $\rightarrow$, $\leftrightarrow$, quantifiers $\exists$, $\forall$, equality symbol " $=$ ", parenthesis "(" and ")", and variables.

