# OpenReview forum: "Logical view on fairness of a binary classification task"
_ICLR.cc/2023/Conference — Submitted to ICLR 2023_

### Official Review · Reviewer_ExFJ · 2022-10-17

**Confidence:** 1
**Clarity, Quality, Novelty And Reproducibility:** NA
**Correctness:** 1
**Technical Novelty And Significance:** 1
**Empirical Novelty And Significance:** Not applicable
**Recommendation:** 1

**Strength And Weaknesses:**

NA

**Summary Of The Paper:**

I am unable to review this paper.

**Summary Of The Review:**

NA

---

### Official Review · Reviewer_ZpSj · 2022-10-23

**Confidence:** 2
**Correctness:** 4
**Technical Novelty And Significance:** 3
**Empirical Novelty And Significance:** Not applicable
**Recommendation:** 5

**Clarity, Quality, Novelty And Reproducibility:**

To me, the paper wasn’t clear instantly. Some may have to do with the structure with the paper, and a lot will have to do with the fact that these parts of mathematics have been a few years for me. But some parts could use some further mathematical explanation. For example, the definition of the random graph in section 1.2 was a bit vague to me. It followed that I couldn’t easily reproduce all the proofs. I had to think on some proofs for a while. I do think there is a novelty in this paper as I’ve never encountered a logical view on fairness. Also, after doing brief research, I couldn’t find any substantial papers on mathematical logic and fairness in machine learning. Furthermore, the paper is based on few references and so it may be that there’s not much to find.

**Strength And Weaknesses:**

Strengths:

Use of first-order logic in machine learning
Paper with profound mathematics with mainly known-results

Weaknesses:

Fields of mathematics are mentioned, but I doubt that they are immediately obvious to regular machine learning researcher. It seems to be model theory at first, next I’m thinking it’s more real analysis. This may be due to the order of the paper. However, many definitions are introduced in Section 1.1.

Throughout the paper and its proofs, I am missing the connection to a more intuitive definition of fairness in a classification task.

**Summary Of The Paper:**

In the development of social sciences to investigate human societies, measurement models in the social sciences did not keep in mind the deep societal reach of  algorithms. This calls for new methods to measure feasible information. However, this paper shows that innovative approaches to obtaining such fair and trustworthy information are heuristic at best.

The ineffectiveness of fairness in a machine learning task is proven by a purely logical view on fairness of a binary classification task. Namely, it is shown that first-order logic language 𝐿 and 𝐿- theory 𝑇 for binary classification is not expressible in the first-order logic 𝐿-formula. This means in first-order logic that fairness and trustworthiness is algorithmically undecidable for binary classification. And so we are unable to find an effective algorithm that decides whether a well-defined formula in logic theory is true. To prove this undecidability, the paper looks at a special class of classifiers that, in logic, are similar to binary classifiers. These classifiers are called random binary classifiers, and they are based on a random graph structure. It is shown that random graphs are isomorphic, which results in the theory of random graphs to be almost surely decidable. However, in a considerable proof it is shown that despite random binary classifiers to be almost surely decidable, one cannot use any first-order logic expression to compare two binary classifiers fairly. Hence, we arrive to the conclusion of the paper, that fairness in binary classification remains to be heuristic at best. The paper suggests further research on higher-order logic.

**Summary Of The Review:**

Overall, I think this is an interesting and possibly, a novel paper. It took me some effort to understand everything, which will probably have to do with me, but maybe also with the way the paper is written.

---

### Official Review · Reviewer_isj2 · 2022-10-26

**Confidence:** 3
**Correctness:** 2
**Technical Novelty And Significance:** 1
**Empirical Novelty And Significance:** Not applicable
**Recommendation:** 3

**Clarity, Quality, Novelty And Reproducibility:**

The paper is unclear about what its contribution is and why that contribution matters. From what I do understand, it doesn't seem novel. There are no experiments.

**Strength And Weaknesses:**

I am not entirely sure what the point of this paper is, unfortunately. The introduction makes claims about fairness and trustworthiness not being decidable, but the preliminaries (which claim to be self-contained) do not provide a formal mathematical definition of either concept that would be necessary to make claims about decidability. This is not made clearer in section 2.1, where the first result concerning inexpressiveness of fairness is claimed. Some translation here between any common fairness metric and the claim here would be helpful. Moreover, I do not understand how the claim in section 3 around undecidability is novel (even if it is valid), which also seems acknowledged in footnote 2. I also do not understand how these claims relate to trust, as the concept does not come up again in the paper. Overall, I am not sure what the contribution of the paper is, or what I should take from it.

**Summary Of The Paper:**

This paper claims that it proves that fairness and trustworthy are undecidable for binary classification. The authors frame this as a first-order logic problem to guide their proof.

**Summary Of The Review:**

See above. Paper's contributions are not clear, both in a big picture sense and in terms of how it relates to the field on algorithmic fairness. It is also not clear what is novel here, even if it were relevant (e.g., see footnote 2 in the paper).

---

### Decision · Program_Chairs · 2023-01-20

**Decision:**

Reject

**Justification For Why Not Higher Score:**

The paper is completely unreadable in its current form.

**Justification For Why Not Lower Score:**

N/A

**Metareview: Summary, Strengths And Weaknesses:**

The paper argues that achieving fairness is an undecidable task using a logical view of binary classification. None of the two reviewers who provided a review were in support of accepting the paper. As an AC who has published multiple papers in the fairness literature, I also read the paper and I share the same concerns as both reviewers.

In its current form, the paper is completely unreadable -- it is not a matter of proof reading the paper but to extensively restructure and rewrite the paper. Moreover, it contains a lot of bold claims that are questionable, e.g., any notion of trustworthiness builds upon fairness, any fairness notions requires comparing the losses two different classifiers achieve. In general, those claims are not supported by the extensive literature of fairness and trustworthiness in machine learning. The response by the authors did not clear out these concerns. I would encourage the authors to revise their paper to engage with the existing literature (the paper cites less than 10 papers), tone down the bold claims and/or support them in a precise, meaningful manner, and think on ways of explaining the material more clearly.